# Risk Factors for Immune Checkpoint Inhibitor-Induced Liver Injury and the Significance of Liver Biopsy

**DOI:** 10.3390/diagnostics14080815

**Published:** 2024-04-14

**Authors:** Miki Kawano, Yoshihiko Yano, Atsushi Yamamoto, Eiichiro Yasutomi, Yuta Inoue, Jun Kitadai, Ryutaro Yoshida, Takanori Matsuura, Yuuki Shiomi, Yoshihide Ueda, Yuzo Kodama

**Affiliations:** 1Division of Gastroenterology, Department of Internal Medicine, Kobe University Graduate School of Medicine, Kobe 650-0017, Japan; kawanom@med.kobe-u.ac.jp (M.K.); yama1004@med.kobe-u.ac.jp (A.Y.); yuta9130@med.kobe-u.ac.jp (Y.I.); jkitadai@med.kobe-u.ac.jp (J.K.); yoshidar@med.kobe-u.ac.jp (R.Y.); tmatsu@med.kobe-u.ac.jp (T.M.); yshiomi@med.kobe-u.ac.jp (Y.S.); yueda@med.kobe-u.ac.jp (Y.U.); kodama@med.kobe-u.ac.jp (Y.K.); 2Department of Gastroenterology, Hyogo Prefectural Kakogawa Medical Center, Kakogawa 675-8555, Japan; eiichiro_yasutomi@yahoo.co.jp

**Keywords:** immune checkpoint inhibitor, liver injury, DILI, histology

## Abstract

Immune checkpoint inhibitor (ICI)-induced liver injury (LI) is a common adverse event, but the clinical characteristics based on the classification of hepatocellular injury and cholestatic types are not fully evaluated. This study aims to analyze risk factors and histological findings in relation to the classification of ICI-induced LI. In total, 254 ICI-induced LI patients among 1086 treated with ICIs between September 2014 and March 2022 were classified according to the diagnostic criteria for drug-induced LI (DILI), and their risk factors and outcomes were evaluated. Kaplan–Meier analyses showed that overall survival in patients with hepatocellular-injury-type LI was significantly longer than others (*p* < 0.05). Regarding pre-treatment factors, the lymphocyte count was significantly higher in patients with ICI-induced LI, especially in hepatocellular-injury-type LI. Gamma glutamyl transferase (γGTP) and alkaline phosphatase (ALP) were also significantly lower in patients with ICI-induced LI (*p* < 0.05). Multivariate analyses revealed that malignant melanoma, high lymphocyte count, and low ALP levels were extracted as factors contributing to hepatocellular-injury-type LI. The histological findings among 37 patients diagnosed as ICI-induced LI via liver biopsy also revealed that the spotty/focal necrosis was significantly frequent in hepatocellular-injury-type LI, whereas ductular reactions were frequently observed in cholestatic-type LI. It is suggested that the histological inflammation pattern in patients with LI is closely correlated with the type of DILI.

## 1. Introduction

Although immune checkpoint inhibitors contribute to the prolongation of survival with respect to many kinds of cancer, they may cause immune checkpoint inhibitor (ICI)-induced liver injury (LI) due to the poor control of immunological mechanisms [1,2]. T-cell activation via the suppression of the Treg function is considered to be involved in the pathology of ICI-induced LI, but direct effects of the enhancement of humoral autoimmunity via autoantibodies existing from before treatment, the increase in cytokines, and injury via complements are also considered to be involved [3,4]. ICI-induced LI is a frequent adverse event and may make the continuation of treatment difficult or lead to liver failure. The prediction of ICI-induced LI may contribute to improvements in patient outcomes and the quality of life. While many predictive factors of ICI-induced LI have been proposed, there is as yet no consensus. 

LI due to cancer is often diagnosed by oncologists using the diagnostic criteria of Common Terminology Criteria for Adverse Events (CTCAEs). ICI-induced LI is treated using corticosteroids and Mycophenolate mofetil, and they are recommended in treatment guidelines as standard treatments [5,6]. However, liver injuries associated with drug therapies are often diagnosed and treated, and therapeutic responses are predicted by hepatologists according to the diagnostic criteria for DILI. According to the diagnostic criteria for DILI, ICI-induced LI is classified into hepatocellular injury and cholestatic types, and the clinical course and treatment response differ according to the disease type. While the pathological findings of ICI-induced liver injuries are heterogeneous, lobular hepatitis primarily affecting the centrilobular region and granulomatous hepatitis are the most common, and lymphocytic cholangitis accompanied by bile duct degeneration or proliferation has also been reported.

In this study, we evaluated the pre-treatment risk factors of ICI-induced LI in real clinical situations at our hospital and compared the disease types of DILI with pathological findings.

## 2. Subjects and Methods

### 2.1. Accumulation of Cases

All cases treated with ICIs at Kobe University Hospital between September 2014 and March 2022 were studied. The patient backgrounds at the beginning of ICI administration and the results of various tests from the beginning of the administration to the patient’s death, 6 months after the last administration of the drug, or November 2019 were retrospectively collected from the clinical records. The items analyzed were the drugs used at the beginning of the administration, type of cancer treated, age, sex, body mass index (BMI), Eastern Cooperative Oncology Group performance status (ECOG PS), drinking history, blood test results, and imaging study results. Concerning the imaging study results, the presence or absence of intrahepatic metastases or hepatic infiltration was also checked.

### 2.2. Diagnostic Criteria

The presence or absence of liver injuries was determined according to CTCAE ver5.0. Patients who showed grade 1 or severe aspartate aminotransferase (AST) elevation according to CTCAE ver5.0 or alanine aminotransferase (ALT) elevation were classified in the LI group. Those with LI with an AST or ALT level exceeding 3 times the upper limit of the institutional standards at the beginning of treatment were excluded from analyses. In addition, patients judged from the clinical course or various test results to have LI due to bacterial cholangitis, drug-induced LI due to other drugs, LI due to circulatory impairment, or LI due to the progression of the primary disease were also excluded from analyses.

Next, when ICI-induced LI was diagnosed based on the diagnostic criteria, the condition was classified according to the diagnostic criteria of DILI into hepatocellular injury, cholestatic, or mixed [7]. ICI-induced LI with an ALT level 5 times the upper limit of the normal range or higher and an ALT/ALP ratio of 5 or higher was classified as the hepatocellular injury type; ICI-induced LI with an ALP level 2 times the upper limit of the normal range or higher and an ALT/ALP ratio of 2 or less was classified as the cholestatic type; ICI-induced LI with an ALT level of 3 times the upper limit of the normal range or higher, an ALP level of 2 times the upper limit of the normal range or higher, and an ALT/ALP ratio of 2 or higher and less than 5 was classified as the mixed type.

Then, concerning the 254 patients who underwent liver biopsy, the presence or absence of inflammation of the liver parenchyma was examined pathologically, and the presence or absence of portal and periportal inflammation was examined histologically.

### 2.3. Statistical Analysis

Fisher’s exact test was performed in the ICI-induced LI group and no LI group using SPSS ver 25.0 at the *p* < 0.05 level of significance, and items that can be risk factors for LI were extracted. Since 21 patients were classified in the LI group, variables were eliminated stepwise from the extracted items, and nominal logistic analysis was performed using the presence/absence of LI as the objective variable. An additional evaluation was carried out in more detail concerning the identified items. 

## 3. Results

### 3.1. Patient Background

The subjects were 1086 patients (818 males and 268 females, with a median age of 70 (21–91) years: 355 with lung cancer, 283 with urethral cancer, 157 with head and neck cancer, 92 with malignant melanoma, 93 with gastrointestinal cancer, 59 with hepatocellular cancer, and 46 with others; 748 treated with anti-programmed death receptor-1 (PD-1) antibody, 253 treated with anti-programmed cell death ligand-1 (PDL-1) antibody, 13 treated with anti-cytotoxic T-lymphocyte-associated protein-4 (CTLA-4) antibody, and 72 treated via combination therapy). All patients who received combination therapy were treated with PD-1 and CTLA-4 antibodies. Of the 1086 patients, 500 showed elevations of liver enzyme levels during the observation period, and 254 who remained after the exclusion of hepatic infiltration or the metastasis of the tumor, bacterial cholangitis, and drug-induced LI were judged to have ICI-induced LI (Figure 1).

Table 1 shows the results of the classification of 254 patients with ICI-induced LI according to CTCAE. ICI-induced LI occurred in 183 patients (24.5%) with the PD-1 antibody, 46 patients (18.2%) with PDL-1, 3 patients (23.1%) with the CTLA-4 antibody, and 21 patients (29.2%) with combination therapy, with no differences in the frequency of occurrence. The treatment period was long in grade 1 and grade 2 cases, and the overall response rate (ORR) also tended to be high in grade 1 and grade 2 patients. ICI-induced LI was treated with steroids in a high percentage of patients: 58% of grade 3 and 59% of grade 4 patients. Figure 2 shows the results of the classification of ICI-induced LI patients based on CTACE by the disease type based on the criteria of DILI. Of the patients with grade 1 ICI-induced LI, 58 had hepatocellular-injury-type LI, and 52 had cholestatic-type LI, but the percentage of patients with cholestatic-type LI decreased with the progression of grade, and in grade 4 ICI-induced LI, there were 34 patients with the hepatocellular injury type compared with only 2 patients with the cholestatic type.

### 3.2. Clinical Characteristics of Patients with ICI-Induced LI

When patients who had ICI-induced LI and those who did not were compared, malignant melanoma was observed more frequently and hepatocellular carcinoma was observed less frequently in the former group. Regarding pre-treatment factors, the lymphocyte count was significantly higher (mean: 1534 vs. 1266, *p* = 0.001), and γGTP and ALP were significantly lower in those with ICI-induced LI (*p* < 0.05). Also, the ICI treatment period was longer (mean 243 days vs. 179 days, *p* = 0.009), and the disease control rate (DCR) was higher (67% vs. 55%, *p* < 0.001) in patients with ICI-induced LI than in those with no ICI-induced LI (Table 2). 

Next, when the hepatocellular-injury-type LI was compared with cholestatic and mixed type LI, among the factors before ICI treatment, the lymphocyte count was significantly higher (mean, 1534 vs. 1274, *p* = 0.003), and ALP and γGTP were lower in the hepatocellular injury type (*p* < 0.005) (Table 3). Concerning factors that lead to the hepatocellular-injury-type LI, malignant melanoma, high pre-treatment lymphocyte count, and low ALP were observed via multivariate analysis (Table 4). The log-rank test of the survival showed no differences between patients with the hepatocellular-injury-type -LI and those with no ICI-induced LI but a significant prolongation of survival times in patients with the hepatocellular-injury-type -LI compared with other patients (*p* = 0.034) (Figure 3).

### 3.3. Type of ICI-Induced LI and Pathological Findings

A liver biopsy was performed for the diagnosis of LI during ICI treatment in 42 patients, of whom 5 had bacterial cholangitis and the hepatic infiltration of malignant tumors. The remaining 37 patients were diagnosed with ICI-induced LI. According to the classification of DILI, the hepatocellular injury type, mixed type, and cholecystic type accounted for 17, 18, and 2, respectively. A histological study was carried out regarding the presence or absence of lobular hepatitis, panlobular hepatitis, centrilobular hepatitis, and parenchymal necrosis, including the appearance of pigmented macrophage, granuloma, councilman body, ballooning, and emperiporesis in the liver parenchyma; and regarding the presence or absence of portal hepatitis, interface hepatitis, fibrosis of portal tract, eosinophil infiltration, ductular reaction, and lymphocytic cholangitis in portal/periportal inflammation (Figure 4A,B).

Inflammation of the liver parenchyma (76%) and spotty/focal necrosis (82%) in hepatocellular-injury-type LI was significantly more frequent than those in mixed type LI (*p* = 0.005 and *p* = 0.020, respectively). However, portal/periportal inflammation (78%) and ductular reaction (44%) were highly observed in mixed-type LI, compared with the hepatocellular injury type (*p* = 0.002 and *p* = 0.032, respectively). (Table 5). In some cases of cholestatic-type LI with jaundice, cholestasis was observed in interlobular bile ducts. In many cases with lymphocytic infiltration, CD8-positive T lymphocytes were observed (Figure 4C).

## 4. Discussion

ICI-induced LI is a relatively common adverse drug reaction. It is treated primarily according to CTACE, and steroids are recommended for grade 3 or more advanced conditions. In our present study, early steroid therapy was also performed for grade 3 or more advanced diseases, with improvements in all patients. However, the concept of liver injuries, including those associated with cancer chemotherapy, is conventionally established as DILI. It is diagnosed according to criteria based on the Roussel Uclaf causality assessment method (RUCAM), but the consensus from the Innovation and Quality DILI Immunotherapy Working Group does not recommend the RUCAM diagnostic criteria for DILI induced by immunotherapy. This is because the scoring of the categories of re-administration items is difficult since the safety of re-administration has not been established [8]. Generally, DILI is classified into hepatocellular injury, cholestatic, and mixed types [9,10]. Recently, disease typing has also been considered important in ICI-induced LI, and steroid therapy has been reported to be effective in hepatocellular-injury-type LI [11]. In our present study, survival-prolonging effects were also observed, particularly in hepatocellular-injury-type LI, suggesting that there are differences in its treatment response and prognosis compared with cholestatic-type LI.

Factors related to ICI-induced LI vary among reports, and it is reported to more frequently affect Asians by race [12] and females by sex [13,14]. Although there are a number of reports denying sex differences in the incidence of DILI [15,16], females were suggested to be more prone to severe DILI [17]. There is no clear sex difference in the incidence of ICI-induced LI, and in this study, while it was observed slightly more frequently in females (25.7%, 69/268) than in males (22.6%, 186/823), the difference was not significant. Recently, a meta-analysis concerning ICI-induced LI has been reported, and being Asian and having a high baseline liver enzyme level are suggested to be related to ICI-induced LI [12]. Concerning treatment, the incidence of ICI-induced LI is reported to be high in initial treatments by some [14] and in re-treatment by others [13]. Its incidence has also been suggested to be high in PD-1 + CTLA-4 combination therapy [14,18,19,20]. There are also reports that a high pre-administration lymphocyte or eosinophil count is related to ICI-induced LI [21,22], but a high eosinophil count is recently considered a risk factor [20]. In addition, factors including fever after the beginning of treatment [23] and a high neutrophil–lymphocyte ratio (NLR) after treatment are considered to be related to ICI-induced LI. In the present study, low biliary system enzyme levels (ALP and γGTP) and a high lymphocyte count before treatment were extracted as risk factors for hepatocellular injury-type LI. Although many factors have been suggested to be involved in ICI-induced LI, there have been only a small number of reports about risk factors for hepatocellular injury-type LI. Detailed evaluation is considered important for conducting appropriate treatment.

As typical histological features of ICI-induced LI, lobular hepatitis primarily affecting the centrilobular region and granulomatous hepatitis are considered the most frequent, but lymphocytic cholangitis accompanied by bile duct degeneration or proliferation is also reported [24]. While differentiation from autoimmune hepatitis often poses problems, CD8-positive T-cell infiltration is also considered important for the diagnosis of ICI-induced LI [25]. In the present study, the infiltration of CD8-positive T cells was also observed in many cases. Although the significance of liver biopsy in ICI-induced LI is unclear, microscopic bile duct injury is reported to be observed even when there is no clear abnormality during imaging examinations, and liver biopsy is suggested as an examination recommended for aggressive pathological exploration and treatment selection [26]. In our present study, biopsies were performed on 42 patients, all of whom had CTCAE grade 3 or more advanced LI. Three of them were diagnosed with cancer infiltration, and another was diagnosed with bacterial cholangitis rather than ICI-induced LI. Liver biopsy was shown anew to be useful for reliably diagnosing ICI-induced LI. Also, in hepatocellular-injury-type LI, centrilobular hepatitis and spotty/focal necrosis were observed significantly more frequently than in cholestatic ICI-induced LI, and portal/periportal hepatitis and ductular reaction were observed significantly more frequently in patients with cholestatic-type LI, indicating that the classification of DILI also has histological support. LI due to CTLA-4 is considered to be characterized by granulomatous hepatitis, including fibrin ring granulomas and central vein endotheliitis, and LI due to PD-1/PD-L1 is considered to be characterized by lobular hepatitis [2]. In the present study, granuloma was found in 5 of 37 patients (14%), 3 of which were treated with combination therapy, including CTLA-4. On the other hand, there were nine cases of LI with CTLA-4, three of which showed granuloma, supporting previous reports [2]. ICI-induced LI is heterogeneous, and the further accumulation of cases is considered necessary for more detailed clarification of its pathology and the determination of treatment principles.

While ICIs are widely used for the treatment of many cancers, there have been a number of reports that ICI-induced LI is related to the outcome. It has been reported that OS is prolonged when ICI-induced LI occurs in ICI treatment for non-small-cell lung cancer [27], hepatocellular cancer [28,29], and many other malignant tumors [13,20,29]. The occurrence of ICI-induced LI will result in a favorable anti-tumor effect because of the induction of natural immunity. LI is a relatively common ICI-induced LI, and the further accumulation of cases is considered necessary for the establishment of its significance.

This study has a few limitations. First, it is a single-center retrospective observational study biased with respect to the patient background, including the disease and drugs used. Next, the number of cases that could be pathologically evaluated was small. It has been reported that typical ICI-induced LI occurred 6–14 weeks after the beginning of the administration of ICIs [30,31,32,33] and that nearly all cases of ICI-induced LI due to nivolumab occurred less than 32 weeks after the beginning of the administration [34]. In this study, the observation period was set until 6 months after the end of treatment. However, no patient had prolonged LI beyond 6 months or developed liver failure due to LI. Recently, however, late-onset ICI-induced LI has also been reported, and the accumulation of liver injury cases those develop longer period after treatment may become necessary in the future [27,35]. Various prognostic factors of ICI-induced LI have been reported, but appropriate treatment selection is expected to be facilitated as factors are clarified in greater detail by the future accumulation of cases.

## 5. Conclusions

The inflammation pattern of hepatitis clarified via liver biopsies in LI observed during the use of ICIs closely correlated with the DILI disease type. In addition, the lymphocyte count before ICI treatment was suggested to be related to the disease type of LI as well as the occurrence of ICI-induced LI. Moreover, patients with hepatocellular injury type ICI-induced LI exhibited significantly longer survival compared with other patients.

## Figures and Tables

**Figure 1 diagnostics-14-00815-f001:**
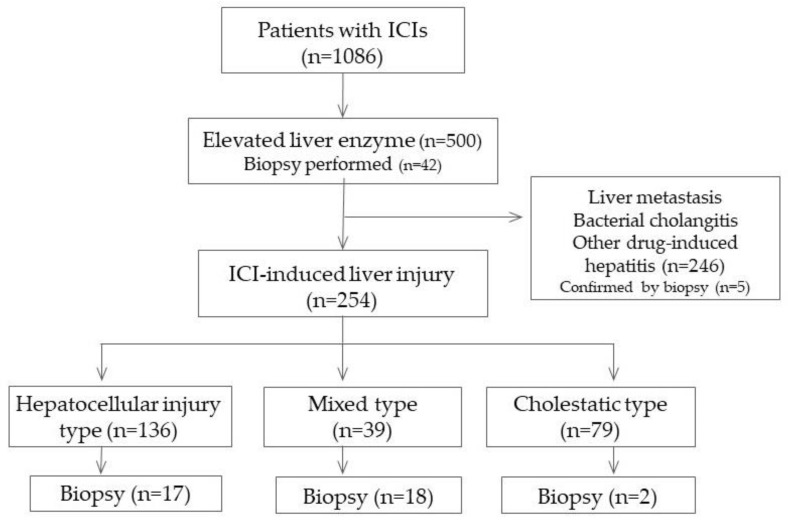
Flowchart of the 1086 patients with hepatic ICI-induced LI included in this study.

**Figure 2 diagnostics-14-00815-f002:**
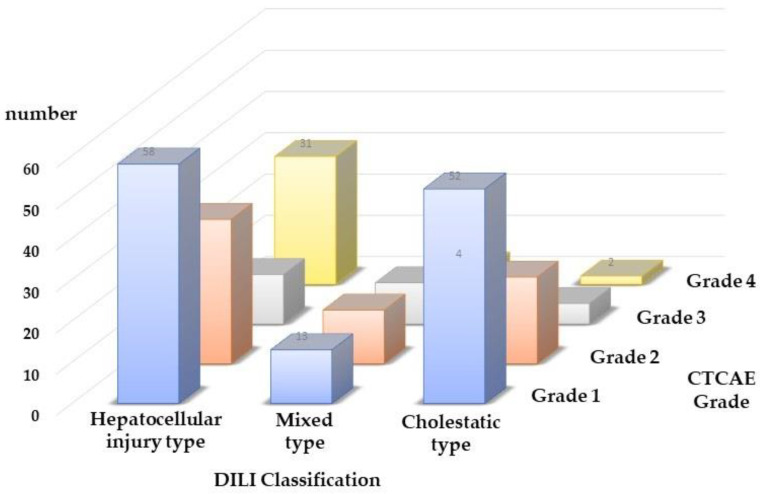
Grading of CTCAE LI by DILI type classification.

**Figure 3 diagnostics-14-00815-f003:**
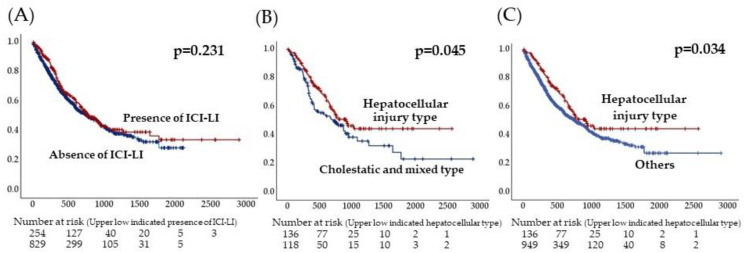
Kaplan–Meier analysis. (**A**) Comparison by the presence/absence of ICI-induced LI. (**B**) Comparison of ICI-induced LI patients by DILI type. (**C**) Comparison of ICI-induced LI patients with hepatocellular injury type and others.

**Figure 4 diagnostics-14-00815-f004:**
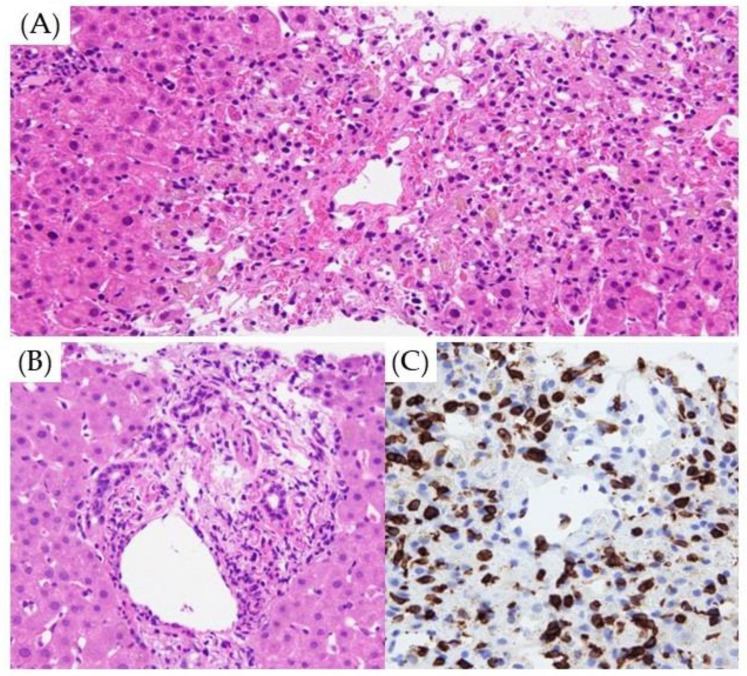
Histopathology of ICI-induced LI. (**A**) Centrilobular hepatitis with endotheliitis and necrosis of hepatocytes is observed around the central vein (Hematoxylin and eosin staining ×100). (**B**) Portal and periportal hepatitis with bile duct degeneration is observed in the portal area (Hematoxylin and eosin staining ×100). (**C**) CD8-positive T lymphocytes are found surrounding the central vein (CD8 immunostaining ×100).

**Table 1 diagnostics-14-00815-t001:** Summary of patients based on CTCAE grading.

	Grade 0 (*n* = 830)	Grade 1 (*n* = 123)	Grade 2 (*n* = 69)	Grade 3 (*n* = 26)	Grade 4 (*n* = 37)
Age	70 (21–89)	69 (35–85)	69 (28–87)	70 (50–82)	69 (30–79)
Sex (male)	631 (76%)	93 (76%)	49 (71%)	19 (76%)	24 (65%)
BMI	22.0 ± 3.5	22.7 ± 4.1	21.9 ± 4.0	21.8 ± 3.7	22.3 ± 4.5
ICI treatment details					
PD-1	562	96	50	15	23
Anti-PDL-1	207	22	12	7	5
Anti-CTLA-4	10	0	1	0	2
Combination	51	5	5	4	7
Laboratory data before therapy					
WBC (/mm^3^)	5950 (2000–67,500)	6000 (2200–14,500)	6250 (2800–19,400)	6650 (3800–17,600)	6600 (3100–14,900)
Lymphocytes (/mm^3^)	1186 (160–5133)	1265 (213–3802)	1396 (216–4462)	1006 (306–2229)	1239 (512–4042)
AST (IU/L)	20 (6–247)	22 (12–56)	22 (11–83)	24 (11–127)	21 (9–76)
ALT (IU/L)	15 (2–221)	16 (6–56)	21 (7–91)	24 (7–180)	18 (8–80)
γGTP (IU/L)	31 (2–1101)	30 (8–277)	32 (10–342)	39.5 (15–154)	36 (12–346)
ALP (IU/L)	89 (10–1391)	84 (10–431)	84 (41–347)	93 (10–376)	99.5 (10–353)
Maximum value					
AST (IU/L)		43.0 (16–80)	84 (31–199)	205 (82–357)	510 (178–1269)
ALT (IU/L)		47.0 (13–78)	101 (22–186)	228 (102–392)	630 (184–2458)
ALP (IU/L)		120.0 (31–823)	131 (35–768)	300 (71–1462)	288 (106–981)
γGTP (IU/L)		67.5 (13–1075)	142 (10–1113)	291 (24–997)	346 (48–1780)
T-Bil (mg/dL)		0.8 (0.2–2.8)	0.9 (0.4–4.5)	1.0 (0.4–3.1)	1.5 (0.6–14.3)
Treatment period (days)	85 (1–1714)	159 (1–2675)	148 (1–1034)	86.5 (1–2024)	43 (1–1018)
ORR	271 (35%)	52 (43%)	31 (47%)	9 (35%)	7 (19%)
DCR	421 (55%)	83 (69%)	44 (67%)	13 (50%)	26 (70%)
Steroid therapy		14 (11%)	9 (13%)	15 (58%)	22 (59%)

CTCAE, common terminology criteria for adverse events; BMI, body mass index; WBC, white blood cells; AST, aspartate aminotransferase; ALT, alanine aminotransferase; ALP, alkaline phosphatase; γGTP, gamma glutamyl transferase; T-Bil, total-bilirubin; ORR, overall response rate; DCR, disease control rate

**Table 2 diagnostics-14-00815-t002:** Comparison of patients with ICI-induced liver injury presence and absence.

	ICI-LI Presence (*n* = 254)	ICI-LI Absence (*n* = 830)	*p*
Age (years)	69 (28–87)	70 (21–91)	0.226
Sex (male/female)	185/69	631/199	0.313
PS	0.69 ± 0.67	0.76 ± 0.67	0.150
BMI (kg/m^2^)	22.3 ± 4.1	22.0 ± 3.5	0.311
Lung cancer	84	274	0.901
Urethral cancer	71	208	0.363
Cervical cancer	41	116	0.494
Malignant melanoma	33	58	0.019 *
Esophageal cancer	8	42	0.148
Intestinal cancer	10	42	0.401
Hepatocellular carcinoma	5	50	0.002 *
Liver involvement	26 (19%)	158 (19%)	0.849
ICI treatment details			
PD-1	183 (72%)	564 (68%)	0.146
Anti-PDL-1	46 (18%)	208 (25%)	0.017 *
Anti-CTLA-4	25 (1%)	83 (1%)	0.976
Combination	20 (8%)	50 (6%)	0.270
Laboratory data before therapy			
WBC (/mm^3^)	6400 (2200–19,400)	5950 (2000–67,500)	0.383
Neutrocyte count (/mm^3^)	4216 (795–15,714)	3946 (288–55,350)	0.580
Lymphocyte count (/mm^3^)	1290 (213–4462)	1186 (160–5133)	0.010 *
NLR	3.19 (0.64–31.43)	3.30 (0.19–43.50)	0.251
AST (IU/L)	22 (9–127)	20 (6–247)	0.194
ALT (IU/L)	18 (6–180)	15 (2–221)	0.241
ALP (IU/L)	88 (10–431)	89 (12–1391)	0.018 *
γGTP (IU/L)	32 (8–346)	31 (2–1101)	0.003 *
CRP (mg/dL)	0.46 (0.1–17.3)	0.43 (0.1–27.7)	0.145
ORR	102 (40%)	291 (35%)	0.204
DCR	170 (67%)	457 (55%)	<0.001 *
ICI treatment period (days)	131 (1–2675)	85 (1–1714)	0.009

ICI, immune checkpoint inhibitor; LI, liver injury; PS, performance status; BMI, body mass index; WBC, white blood cells; NLR, neutrophil–lymphocyte ratio; AST, aspartate aminotransferase; ALT, alanine aminotransferase; ALP, alkaline phosphatase; γGTP, gamma glutamyl transferase; CRP, C reactive protein; ORR, overall response rate; DCR, disease control rate; * *p* < 0.05.

**Table 3 diagnostics-14-00815-t003:** Comparison of patients with ICI-induced liver injury by DILI type.

	Hepatocellular Injury Type (*n* = 136)	Cholestatic and Mixed Type (*n* = 118)	*p*
Age (years)	69 (35–87)	69.5 (28–85)	0.866
Sex (male/female)	98/38	87/31	0.766
BMI (kg/m^2^)	22.4 ± 4.1	22.1 ± 4.1	0.339
ICI treatment details			
PD-1	102 (75%)	81 (69%)	0.331
Anti-PDL-1	22 (16%)	24 (20%)	0.395
Anti-CTLA-4	1 (1%)	12 (1%)	0.642
Combination	10 (7%)	11 (9%)	0.575
Liver involvement	26 (19%)	21 (18%)	0.787
Laboratory data before therapy			
WBC (/mm^3^)	6450 (2700–19,400)	6250 (2200–14,500)	0.026 *
Neutrocyte count (/mm^3^)	4216 (1290–15,714)	4234 (795–11,310)	0.181
Lymphocyte count (/mm^3^)	1334 (213–4462)	1247 (216–3165)	0.003 *
NLR	3.01 (0.81–12.00)	3.46 (0.64–31.43)	0.177
AST (IU/L)	21 (10–127)	23 (9–83)	0.836
ALT (IU/L)	17 (6–180)	18 (7–91)	0.750
ALP (IU/L)	80 (10–210)	98 (12–431)	<0.001 *
γGTP (IU/L)	30 (10–346)	37 (8–342)	0.015 *
Maximum value			
AST (IU/L)	64 (16–1269)	62 (22–509)	<0.001 *
ALT (IU/L)	86 (26–2458)	64 (13–836)	<0.001 *
ALP (IU/L)	104 (31–648)	221 (113–1462)	<0.001 *
γGTP (IU/L)	85 (10–1780)	167 (22–1113)	<0.001 *
ORR	57 (42%)	44 (37%)	0.419
DCR	94 (69%)	76 (64%)	0.372
ICI treatment period (days)	128 (1–2185)	134 (1–2675)	0.664
Steroid therapy	35 (26%)	25 (21%)	0.394
Treatment continued/re-treatment	97 (71%)	87 (74%)	0.673

ICI, immune checkpoint inhibitor; DILI, drug-induced liver injury; BMI, body mass index; WBC, white blood cells; NLR, neutrophil–lymphocyte ratio; AST, aspartate aminotransferase; ALT, alanine aminotransferase; ALP, alkaline phosphatase; γGTP, gamma glutamyl transferase; ORR, overall response rate; DCR, disease control rate; * *p* < 0.05.

**Table 4 diagnostics-14-00815-t004:** Multivariate analysis in relation to the presence of ICI-LI with hepatocellular injury type.

	*p*	HR	95%CI
Malignant melanoma	0.041 *	1.945	1.029–3.677
Liver cancer	0.061		
Anti-PD-1 antibody	0.163		
ALP	0.007 *	0.991	0.984–0.997
γGTP	0.056		
Lymphocyte count	0.013 *	1.000	1.000–1.001

ICI, immune checkpoint inhibitor; LI, liver injury; ALP, alkaline phosphatase; γGTP, gamma glutamyl transferase; HR, hazard ratio; CI, confidential intervals * *p* < 0.05.

**Table 5 diagnostics-14-00815-t005:** Relationship between histological findings and DILI classification.

Histological Findings	DILI Classification			*p*
	Hepatocellular Injury Type (*n* = 17)	Mixed Type (*n* = 18)	Cholestatic Type (*n* = 2)	
Centrilobular hepatitis	13 (76%) *	5 (28%) *	0 (0%)	0.005 *
Spotty/focal necrosis	14 (82%) *	8 (44%) *	2 (100%)	0.020 *
Pigmented macrophage	6 (35%)	4 (22%)	0 (0%)	n.s.
Granuloma	1 (6%)	4 (22%)	0 (0%)	n.s.
Councilman body	9 (53%)	5 (28%)	1 (50%)	n.s.
Ballooning	5 (29%)	3 (17%)	0 (0%)	n.s.
Emperiporesis	5 (29%)	3 (17%)	0 (0%)	n.s.
Interface hepatitis	4 (24%) *	11 (61%) *	2 (100%)	0.028 *
Portal inflammation	4 (24%) *	14 (78%) *	2 (100%)	0.002 *
Fibrosis of portal tract	5 (29%)	6 (33%)	0 (0%)	n.s.
Eosinophil infiltration	6 (35%)	9 (50%)	2 (100%)	n.s.
Ductular reaction	2 (12%) *	8 (44%) *	2 (100%)	0.032 *

DILI, drug-induced liver injury; n.s., not significant; * *p* < 0.05.

## Data Availability

Data are contained within the manuscript.

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
