# Peer review of "Risk Factors for Immune Checkpoint Inhibitor-Induced Liver Injury and the Significance of Liver Biopsy"

_diagnostics, 2024, doi:10.3390/diagnostics14080815_

Round 1

Reviewer 1 Report

Comments and Suggestions for Authors

The authors have investigated the ICI-induced liver injury by clinical and pathological aspects. Although the focus of study is interesting, I have several comments mainly pathological assessments.

Although the authors have classified DILI into hepatocellular, cholestatic, or mixed type, it is more better classifying it into hepatocellular-injury type, cholestatic type, or mixed type (Original usage Ref.#7 are hepatocellular injury, cholestatic liver injury, and mixed liver injury).

In pathological examination, please reconsider following matters:

1. How about the degree of necrosis/injury of hepatocytes? Is there difference between hepatocellular-injury type and cholestatic type? Possible factors which are pathologically evaluated are the number of Councilman bodies, ballooning degeneration, and the figures of emperipolesis.

2. How about the status of interface hepatitis and fibrosis of the portal tracts? Is there difference between hepatocellular-injury type and cholestatic type?

3. How about the type of cholestasis? bile canaliculi? or interlobular bile ducts? Is there difference between hepatocellular-injury type and cholestatic type?

4. The figures of non-suppurative destructive cholangitis or vanishing bile ducts are observed?

5. Infiltration of CD8-positive T-lymphocytes should be assessed in biopsy specimens.

Author Response

Dear Reviewer

Thank you very much for your important comment for our manuscript. According to your comment, we modified and updated our manuscript. I hope revised manuscript is more clinically more meaningful.

1. Although the authors have classified DILI into hepatocellular, cholestatic, or mixed type, it is more better classifying it into hepatocellular-injury type, cholestatic type, or mixed type (Original usage Ref.#7 are hepatocellular injury, cholestatic liver injury, and mixed liver injury).

[Reply] Thank you for your notification. According to the comment, I corrected from “hepatocellular type” to “hepatocellular injury type”.

2. How about the degree of necrosis/injury of hepatocytes? Is there difference between hepatocellular-injury type and cholestatic type? Possible factors which are pathologically evaluated are the number of Councilman bodies, ballooning degeneration, and the figures of emperipolesis.

[Reply] We evaluated pathologically 42 biopsied samples and summarized in table 5. Histological findings including councilman bodies, ballooning degeneration, emperiporesis as well as pigmented macrophage, granuloma were also evaluated and summarized. In addition, the significance of granuloma in relation to CTLA-4 was also included in the discussion.

3. How about the status of interface hepatitis and fibrosis of the portal tracts? Is there difference between hepatocellular-injury type and cholestatic type?

[Reply] Interface hepatitis and fibrosis were also evaluated and summarized in table 5. Interface hepatitis and fibrosis were more frequent in mixed type compared with hepatocellular injury type.

4. How about the type of cholestasis? bile canaliculi? or interlobular bile ducts? Is there difference between hepatocellular-injury type and cholestatic type?

[Reply] Cholestasis was sometimes seen in cases of jaundice, but most of the cases were interlobular bile ducts. I added this finding to the result section.

5. The figures of non-suppurative destructive cholangitis or vanishing bile ducts are observed?

[Reply] We excluded 2 cases bacterial cholangitis. However, no case was histologically confirmed as vanishing bile ducts. Ductular reaction was frequently found in mixed type (table 4).

6. Infiltration of CD8-positive T-lymphocytes should be assessed in biopsy specimens.

[Reply] As reviewer pointed out, CD8-positive T lymphocytes were common in this study and the histological finding was added in Figure 4C.

Reviewer 2 Report

Comments and Suggestions for Authors

The study investigates Immune Checkpoint Inhibitor-induced Liver Injury (ICI-induced LI) in a large cohort of cancer patients. It offers interesting observations about correlations between LI type, pre-treatment factors, histological findings, and patient outcomes. However, certain aspects could be strengthened to increase the study's impact.

·       Author stated that “748 treated with anti-programmed death receptor-1 (PD-1) antibody, 253 101 treated with anti-programmed cell death ligand-1 (PDL-1) antibody, 13 treated with anti-102 cytotoxic T-lymphocyte-associated protein-4 (CTLA-4) antibody, and 72 treated by com-103 bination therapy)”. The author should explain the different treatment regimes and which therapy group finds the higher number of Liver Injury.

·       Did the author have the data of the patient’s follow-up of more the 6 months 

Author Response

Dear Reviewer

Thank you very much for your important comment for our manuscript. According to your comment, we modified and updated our manuscript. I hope revised manuscript is more clinically more meaningful.

1. Author stated that “748 treated with anti-programmed death receptor-1 (PD-1) antibody, 253 101 treated with anti-programmed cell death ligand-1 (PDL-1) antibody, 13 treated with anti-102 cytotoxic T-lymphocyte-associated protein-4 (CTLA-4) antibody, and 72 treated by com-103 bination therapy)”. The author should explain the different treatment regimes and which therapy group finds the higher number of Liver Injury.

Response to reviewer’s comment

We reviewed the case again and added the following statement to result section. “ICI-induced LI occurred in 183 patients (24.5%) with PD-1 antibody, 46 patients (18.2%) with PDL-1, 3 patients (23.1%) with CTLA-4 antibody, and 21 patients (29.2%) with combination therapy, with no difference in frequency of occurrence.”

2. Did the author have the data of the patient’s follow-up of more the 6 months

Response to reviewer’s comment

 Thank you for your comment. Based on the comment, we added the following sentence in the discussion section. “However, no patient had prolonged LI beyond 6 months or developed liver failure due to LI.”

Round 2

Reviewer 1 Report

Comments and Suggestions for Authors

The authors have appropriately responded to the reviewer’s comments and revised the manuscript. The revised version of the manuscript is significantly improved. I have no further comment.